# A Fast and Efficient Approach to Strength Prediction for Carbon/Epoxy Composites with Resin-Missing Defects

**DOI:** 10.3390/polym16060742

**Published:** 2024-03-08

**Authors:** Hongfeng Li, Feng Li, Lingxue Zhu

**Affiliations:** 1School of Mechanical and Aerospace Engineering, Jilin University, Changchun 130025, China; lhfnjtech@163.com; 2Department of Mathematics, Jinling Institute of Technology, Nanjing 211169, China

**Keywords:** carbon/epoxy composites, resin-missing defects, strength prediction method, dimension reduction, Chebyshev polynomials

## Abstract

A novel method is proposed to quickly predict the tensile strength of carbon/epoxy composites with resin-missing defects. The univariate Chebyshev prediction model (UCPM) was developed using the dimension reduction method and Chebyshev polynomials. To enhance the computational efficiency and reduce the manual modeling workload, a parameterization script for the finite element model was established using Python during the model construction process. To validate the model, specimens with different defect sizes were prepared using the vacuum assistant resin infusion (VARI) process, the mechanical properties of the specimens were tested, and the model predictions were analyzed in comparison with the experimental results. Additionally, the impact of the order (second–ninth) on the predictive accuracy of the UCPM was examined, and the performance of the model was evaluated using statistical errors. The results demonstrate that the prediction model has a high prediction accuracy, with a maximum prediction error of 5.20% compared to the experimental results. A low order resulted in underfitting, while increasing the order can improve the prediction accuracy of the UCPM. However, if the order is too high, overfitting may occur, leading to a decrease in the prediction accuracy.

## 1. Introduction

Fiber-reinforced composites are widely used in aerospace, automotive, and wind power generation due to their high specific strength, specific modulus, fatigue strength, good corrosion resistance, and low density [1]. Many composites are transversely isotropic materials, and their failure modes are complex and varied. Although the mechanical properties of the composite structure can be obtained by an experimental method, it requires a high cost and complex preparation process. The finite element method can reduce costs compared to expensive experimental research. However, when calculating the complex structure, if the high-precision model is used directly for analysis, it will not only consume a lot of computing time but also generate huge computing costs. An effective solution is to replace the high-precision model with an approximate model, which achieves a compromise between computational accuracy and computational cost to improve the solution efficiency. The commonly used approximate models include the response surface methodology (RSM), Kriging model, and artificial neural networks (ANNs).

The approximate model, which is based on the RSM, is simple and requires little computation. Saberian et al. [2] created a mathematical model using the RSM to forecast the tensile strength of glass fiber/epoxy composites. The study examined the impact of fiber length, fiber content, and silica nanoparticles on tensile strength. The predicted values were in good agreement with the experimental data. Ghasemi et al. [3,4] applied the RSM to predict and optimize the tensile strength of polypropylene-based nanocomposites. Siddique et al. [5] examined the impact of process variables on the tensile properties of fiber-reinforced composites using the RSM. The Box–Behnken design in the RSM is used as a DOE method to conduct the modeling process. Liu et al. [6] studied the influences of process parameters on the mechanical strength of short-carbon-fiber-reinforced composites by the RSM and gray relational analysis. The reliability of the RSM has been verified through experiments. Srinivasan et al. [7] investigated the prediction of the wear strength of metal matrix composites using the RSM, and a good agreement between the experimental and predicted value was found. He et al. [8] established a response surface quadratic model for a carbon fiber composite hull and studied its strength and stability.

Although increasing the order of the model can improve the prediction accuracy of the RSM, it is not sufficient for dealing with highly nonlinear problems. The Kriging model has the capacity for local estimation, and it is easier to obtain ideal fitting results than when using the RSM when dealing with highly nonlinear problems.

Haeri et al. [9] approximated the mechanical model of composite laminates using an advanced Kriging surrogate model. The accuracy and efficiency of the model were verified by comparing it with the neural network method. Davidson et al. [10] developed a strength prediction model for wavy fiber composites based on the Kriging model to predict the compressive strength. Ameryan et al. [11] investigated the shear strength correlations and reliability assessments of sandwich composite structures by using the Kriging model. The results indicated that the model can make more accurate predictions with a smaller sample size. Zhao et al. [12] applied the Kriging surrogate model to examine the impact of the percentage of various components on the mechanical properties of Poly Ternary Composites. The study found that the Kriging model is a suitable option for predicting the mechanical properties of composites when only limited data are available. This can significantly reduce the experimental costs. Su et al. [13] used the Kriging model to predict the stiffness of an injection-molded short-fiber-reinforced composite. The Kriging model demonstrated high accuracy when compared to the experimental results. Zhou et al. [14] developed an approach based on the Kriging model and the U learning function to predict the mechanical properties of composite structures. The results indicate that the prediction model, which is based on the adaptive Kriging method, has an acceptable level of computational accuracy and can significantly enhance computational efficiency. Keshtegar et al. [15] proposed a buckling load prediction model for composite laminates based on the adaptive Kriging model. The proposed method is highly computationally accurate and significantly reduces computational costs.

The accuracy of the Kriging-based approximation model is closely related to the number of sample points, particularly for large and complex engineering problems and anisotropic high-dimensional problems. The RSM and the Kriging model both have greater difficulty in dealing with problems with many factors and ambiguous information, while the ANN has greater superiority in dealing with such problems.

Kumar et al. [16] established a failure strength prediction model for composite laminates based on radial basis function neural network and generalized regression neural network models and found that the prediction accuracy of the generalized regression neural network model was better than that of the radial basis function neural network model. Hammoudi et al. [17] developed a compressive strength prediction model for recycled concrete based on the ANN and RSM, respectively, and conducted a comparative analysis of the two prediction models. The ANN model showed better accuracy. Zhang et al. [18] predicted the mechanical properties of composite laminate by setting up an ANN model. The results obtained from the ANN model are in good agreement with the finite element analysis, indicating that the model can accurately predict the mechanical properties of composite laminates. Liu et al. [19] examined the impact of crack parameters on the residual compressive strength of composite panels using an ANN. The results show that the ANN model not only has a strong learning ability, it can also quickly and accurately predict the compression strength of a composite laminate. Divya et al. [20,21] used an ANN to forecast the compressive strength of the engineered cementitious composite. Lee et al. [22] employed an ANN to study the effect of different alloying elements on aluminum matrix composites. Zakaulla et al. [23] predicted the mechanical properties of polyetheretherketone composites using an ANN. The ANN was superior in dealing with complex problems but had weaknesses in dealing with small samples.

To address the current issues, we developed a novel univariate Chebyshev prediction model (UCPM). This model combines the dimension reduction method (DRM) with Chebyshev polynomials. A parameterization script for the finite element model was established using Python during the model construction process. A comparison was made between the UCPM and the conventional Chebyshev polynomial fitting model, and the superiority of the UCPM was demonstrated. To validate the model, specimens with different defect sizes were prepared using the VARI process, the mechanical properties of the specimens were tested, and a comparative analysis was conducted of the experimental results. The impact of the fitting order on the accuracy of the UCPM was also examined.

## 2. Problem Statement

Resin-missing is a defect where the resin content of a local area on the surface or inside of composites is less than that of the surrounding area. Figure 1 illustrates the resin-missing defects of composite laminates. The green part represents the fiber, and the gray part represents the resin. Due to the process or improper operation, resin-missing defects are easily produced in the molding process for thick and complex configurations of composite structures. Incomplete air removal may occur when using the VARI process to prepare large or complex composite structures due to premature gelation or an uneven resin flow, resulting in certain areas being devoid of resin, as shown in Figure 1. In severe cases, numerous fiber bundles are not encapsulated by the resin, leaving visible fiber bundles exposed at the macroscopic level. The defects will affect the mechanical properties and service life of the composite.

In practical engineering applications, numerical simulation can predict the strength of composites with resin-missing defects. However, constructing a high-precision finite element model is time-consuming and computationally expensive. To improve the solution efficiency, an approximate model can be constructed instead of a high-precision model, striking a balance between the calculation accuracy and cost. This paper focuses on composite laminates with resin-missing defects, as shown in Figure 2. The size of the composite laminate with a resin-missing defect is 250 × 25 × 1.35 mm^3^. The resin-missing defect is located on the surface of the composite laminate. The defect type is full fiber exposure, and the length is 20 mm. To obtain the strength prediction model of composite laminates with resin-missing defects, the geometric parameters (*L_e_*, *W_e_*, *W_d_*, and *t_d_*), which impact the strength of composite laminates, are chosen as design variables, whereas the strength is treated as the objective function. The mathematical model is established as follows:(1)$$G\left(x\right)=F\left(L_{e},W_{e},W_{d},t_{d}\right)$$
where ***G***(***x***) is the strength, *L_e_* is the distance between the defect and the bottom edge, *W_e_* is the distance between the defect center and the side, *W_d_* is the defect width, and *t_d_* is the defect thickness.

The range of every design variable is chosen by the overall size of the composite laminate, as demonstrated in Table 1.

## 3. Construction of Prediction Models

### 3.1. Chebyshev Polynomial Fitting

Formula (1) shows that the established mathematical model is an implicit function. To obtain the explicit expression of the response function, polynomials are used for fitting. Among these, Chebyshev polynomial fitting is a common function approximation method based on Chebyshev polynomials. Chebyshev polynomials have excellent approximation properties, enabling them to fit various function types. Chebyshev polynomial fitting involves fitting data points by selecting an appropriate Chebyshev polynomial as a basis function and adjusting the polynomial coefficients. The polynomials can be expressed as follows:(2)$$T_{n}\left(x\right)=\cos\left(n\arccos x\right)\;\left|x\right|\leq 1$$

If *x* = cos *θ*, then *T_n_*(*x*) = cos *nθ*, 0 ≤ *θ* ≤ π.

Since the Chebyshev polynomials are defined in [−1,1], the independent variables ***x*** ∈ [a,b] need to be converted to [−1,1], and the transformation formula is presented as follows [24]:(3)$$t_{i}=\frac{2x_{i}-\left(a_{i}+b_{i}\right)}{b_{i}-a_{i}},i=1,2,\ldots ,n$$

***G***(***x***) is a continuous function that can be fitted using the Chebyshev series. The approximate expression is shown as follows:(4)$$G\left(x\right)\approx f\left(t\right)={\sum_{i_{1}=0}^{k}\sum_{i_{2}=0}^{k}\ldots \sum_{i_{n}=0}^{k}\left(\frac{1}{2}\right)}^{p}C_{i_{1}i_{2}\ldots i_{n}}T_{i_{1}i_{2}\ldots i_{n}}\left(t\right)$$
where *k* is the degree of the Chebyshev polynomial, *p* denotes the total number of zeros to occur in the subscript *i*_1_*i*_2_…*i*_n_, and *T_i_*_1*i*2_…*_in_* (***t***) is the n-dimensional Chebyshev polynomial.

$C_{i_{1}i_{2}\ldots i_{n}}$ are the coefficients of the Chebyshev polynomial, which can be computed using the Gauss–Chebyshev integration formula [25]:(5)$$\begin{array}{cc} C_{i_{1}i_{2}\ldots i_{n}} & ={\left(\frac{2}{\pi }\right)}^{n}\int_{-1}^{1}\int_{-1}^{1}\ldots \int_{-1}^{1}\frac{f\left(t\right)T_{i_{1}i_{2}\ldots i_{n}}\left(t\right)}{\sqrt{1-t_{1}^{2}}\sqrt{1-t_{2}^{2}}\ldots \sqrt{1-t_{n}^{2}}}dx_{1}dx_{2}\ldots dx_{n}\\ & \approx {\left(\frac{2}{m}\right)}^{n}\sum_{j_{1}=1}^{m}\sum_{j_{2}=1}^{m}\ldots \sum_{j_{n}=1}^{m}f\left(t_{j_{1}},t_{j_{2}},\ldots ,t_{j_{n}}\right)\;T_{i_{1}i_{2}\ldots i_{n}}\left(t_{j_{1}},t_{j_{2}},\ldots ,t_{j_{n}}\right) \end{array}$$
where *m* denotes the quantity of interpolation points (here, *m* = *k* + 1), *n* describes the number of dimensions, $f\left(t_{j_{1}},t_{j_{2}},\ldots ,t_{j_{n}}\right)$ denotes the response value, and $t_{j_{n}}$ is the root of the (*k* + 1)th-order Chebyshev polynomial:(6)$$t_{j_{n}}=\cos\left(\frac{2j_{n}-1}{k+1}\frac{\pi }{2}\right),\;j_{n}=1,2,\ldots ,k+1$$

Chebyshev polynomial fitting has a wide range of applications in data fitting and function approximation. Due to the approximation nature of Chebyshev polynomials, they provide a better fit and perform well with fewer data points or in the presence of noise. However, when there are many independent variables, fitting the response function directly using Chebyshev polynomials can be computationally expensive.

### 3.2. Dimension Reduction Method (DRM)

The DRM aims to convert a high-dimensional function into the sum of lower-dimensional functions, which reduces the difficulty of the solution and the amount of computation, thus improving the computational efficiency. Consequently, based on the DRM, the response function ***G***(***x***) can be expressed as follows [26,27]:(7)$$\begin{array}{cc} G\left(x\right) & =G_{0}+\sum_{i=1}^{n}G_{i}(x_{i})+\sum_{1\leq i\leq j\leq n}G_{ij}(x_{i},x_{j})+\ldots +\\ & \sum_{1\leq i_{1}<\ldots <i_{k}\leq n}G_{i_{1}i_{2}\ldots i_{k}}(x_{i_{1}},x_{i_{2}},\ldots ,x_{i_{k}})+\ldots +G_{12\ldots n}(x_{1},x_{2},\ldots ,x_{n}) \end{array}$$

According to the univariate DRM [28], the lower-order terms have a greater impact than higher-order terms in continuously differentiable functions, meaning that higher-order terms can be disregarded in calculations. Consequently, the response function ***G***(***x***) can be approximated as follows:(8)$$G\left(x_{1},x_{2},\ldots ,x_{n}\right)≅\sum_{i=1}^{n}G_{i}(x_{i})+G_{0}$$

According to the DRM, each term in Equation (8) can be expanded at a specified point ***μ*** = (*μ*_1_, *μ*_2_,…, *μ_n_*):(9)$$\left\{\begin{array}{l} G_{0}=G(\mu )\\ G_{i}(x_{i})=G\left(\mu_{1},\ldots ,\mu_{i-1},x_{i},\mu_{i+1},\ldots ,\mu_{n}\right)-G_{0} \end{array}\right.$$
where *μ_i_* is the mean of *x_i_*.

### 3.3. Univariate Chebyshev Prediction Model (UCPM)

For a univariate function ***G***(*x_i_*), *x_i_* ∈ [*a_i_*,*b_i_*], we can obtain the approximate function by the Equation (4):(10)$$G\left(x_{i}\right)\approx f\left(t_{i}\right)=\sum_{j=0}^{k}{\left(\frac{1}{2}\right)}^{p}C_{i,j}T_{j}\left(t_{i}\right)$$
where *C_i_*_,*j*_ is the coefficient of the Chebyshev polynomial, and *T_j_*(*t_i_*) is the Chebyshev polynomial containing only the *i*-th variable.

Referring to Equation (5), *C_i_*_,*j*_ can be written as follows:(11)$$C_{i,j}=\frac{2}{m}\sum_{l=1}^{m}f\left(t_{i,l}\right)\;T_{j}\left(t_{i,l}\right)$$

The response values ***f***(*t_i_*_,*l*_) of the interpolation point can be obtained by finite element calculation.

Univariate functional expressions for a single independent variable can be obtained based on integrating Equations (2) and (11) into Equation (10). Then, we can obtain the UCPM using Equation (8).
(12)$$G\left(x_{1},x_{2},\ldots ,x_{n}\right)\approx \sum_{i=1}^{n}\sum_{j=0}^{k}{\left(\frac{1}{2}\right)}^{p}C_{i,j}T_{j}\left(t_{i}\right)-(n-1)G_{0}$$

In comparison to the original Chebyshev polynomial fitting model, the UCPM significantly reduces the computational difficulty and cost. When there are *m* independent variables, *p*-th-order Chebyshev polynomials are used for fitting. The UCPM needs *m* × (*p* + 1) + 1 sample points, whereas the original Chebyshev polynomial fitting model needs (*p* + 1)*^m^* sample points. Figure 3 illustrates the UCPM construction process.

To construct a prediction model based on the UCPM, it is necessary to determine the order of the Chebyshev polynomials. Constructing a predictive model based on the UCPM requires determining the appropriate order of the Chebyshev polynomials. The higher the order is, the more flexible the polynomials are, but too high an order may result in overfitting, while too low an order may result in underfitting. It is important to choose the order carefully to avoid these issues. Therefore, selecting the appropriate order of Chebyshev polynomials can improve the results in data fitting problems. This study selects a 5th-order UCPM to fit the mathematical model and calculates the response of the interpolation points using the finite element method.

### 3.4. Finite Element Analysis

The finite element model was modeled as a 3D solid using ABAQUS, as shown in Figure 4. The size of the composite laminate with a resin-missing defect was 250 × 25 × 1.35 mm^3^. The laminates consisted of six layers, each with a thickness of 0.225 mm, oriented in [0°]_6_. After conducting the mesh independence test, a mesh size of 1 mm was selected, as shown in Figure 5. The element types of the laminates and defects were C3D8R. One end of the laminate was fixed, while the other end applied displacement. The material used was a carbon/epoxy composite, and the related material parameters are detailed in Table 2 and Table 3. The material parameters were obtained by experimental tests. The Brittle Cracking constitutive model was employed to compute the resin-missing defects. A VUMAT subroutine was developed to calculate the strength and failure mode of composite laminates. The subroutine uses the 3D Hashin damage initiation criteria [29] and the Camanho damage evolution criteria [30]. The analysis step utilizes Dynamic Explicit.

Based on Section 3.3, the finite element model needs to be solved repeatedly to establish the prediction model, which is a time-consuming process. To improve the computational efficiency and reduce the manual modeling workload, parametric modeling and batch calculation of the finite element model are carried out using Python 3.6.

## 4. Specimens and Experiments

### 4.1. Specimen Preparation

To verify the validity of the model, composite laminates with varying defect sizes were prepared. Five different percentages of resin-missing defects (5.3%, 8.0%, 10.7%, 13.3%, and 16.7%) were identified. A total of twenty-five specimens were produced, with five specimens for each defect size. The laminates were made of T300 carbon fiber unidirectional cloth and epoxy vinyl ester resin (GE-7118A). The hardener was GE-7118B, and the weight ratio was 100:30. The fiber cloth was supplied by Yixing Zhongtan Technology Inc. (Wuxi, China), and the epoxy resin and hardener were supplied by Wells Advanced Materials Co., Ltd. (Shanghai, China). The curing conditions were one day at 25 °C followed by an 8 h post-cure. A water-soluble release agent (PARTALL FILM #10) was used to prefabricate the resin-missing defect to achieve an absence of resin in the resin-missing area, as shown in Figure 6a. The release agent dries to form a smooth and glossy barrier film that effectively protects the fibers from resin infiltration. Then, the VARI process was used to prepare composite laminates. The preparation process is shown in Figure 6b–e. The prepared composite laminate is depicted in Figure 6f. The composite laminates were cut to the specified dimensions using a water jet cutting machine.

In order to verify the existence of resin-missing defects, scanning electron microscopy (SEM) was used to observe the cross-sections of the defective and non-defective parts of the composite laminates, respectively, and these were compared with the fibers, as shown in Figure 7. As can be seen from Figure 7a, the surface of the non-resin-coated fibers is smooth, and the fibers are loose and non-adhesive, and there are no inclusions between the fibers. As can be seen from Figure 7b, the fiber surface of the resin-missing defect part is also smooth, and the fiber is loose and non-adhesive, and there is no resin on the fiber surface. Compared with the surface of the fiber, it was found that the two are basically the same. It can be seen from Figure 7c that the surface of the fiber without defects is covered by resin, and the fibers are mutually adhered to each other. It can be seen from the above comparative analysis that the preparation of the resin-missing defect was successful.

### 4.2. Experimental Method

The composite laminates were tested for tensile properties using the testing machine (MTS System Corp, Eden Prairie, MN, USA), with reference to ASTM D3039 [31]. One end of the specimen was fixed, and a vertical upward displacement was applied to the other end at a loading rate of 2 mm/min. The frequency of data collection was one point per second. The experiment was stopped upon specimen destruction. Figure 8 illustrates the experimental setup.

## 5. Results and Discussion

### 5.1. Experimental Results

Figure 9 illustrates the experimental results for composite laminates with different resin-missing defect sizes. Figure 8 shows that the experimental curves for composite laminates with varying sizes of resin-missing defects follow a consistent pattern. The curves exhibit a linear increase at the initial stages, followed by a sharp decline upon reaching the ultimate load. The average tensile strengths of laminates with different resin-missing defect sizes were calculated to be 1570.46 MPa, 1567.56 MPa, 1509.19 MPa, 1495.71 MPa, and 1384.72 MPa, respectively.

### 5.2. Finite Element Results

Figure 10 and Table 4 display the results obtained from the FEM. The curves obtained from finite elements are consistent with the experimental curves, as shown in Figure 10. Table 4 indicates that the maximum error between the finite element calculations and experimental results is 3.06%. These results demonstrate that the finite element calculation method is reasonable and feasible and can be used for subsequent response calculations at interpolated points.

### 5.3. Accuracy Analysis of Strength Prediction Model

Table 5 shows a comparison between the predicted and experimental values for the strength of composite laminates with varying sizes of resin-missing defects. The predicted strength of the composite laminate with defects of 5.3%, 8.0%, 10.7%, 13.3%, and 16.7% is 1499.81 MPa, 1490.27 MPa, 1485.93 MPa, 1474.36 MPa, and 1456.79 MPa, respectively. The error with in experiment is 4.50%, 4.93%, 1.54%, 1.43%, and 5.20%, respectively.

The maximum prediction error of the fifth-order UCPM is 5.20%. For one variable, the fifth-order UCPM requires six interpolation points. As the number of interpolation points increases, so does the prediction accuracy. Consequently, raising the number of interpolation points will enhance the prediction accuracy.

### 5.4. Performance Evaluation of UCPMs of Different Orders

Chebyshev polynomials are a type of polynomial that are used to fit functions with specific properties that minimize fitting errors over a given interval. Section 3 shows that the accuracy of the UCPM is related to the number of interpolation points. Therefore, varying the order of the Chebyshev polynomials can improve the accuracy of the UCPM. To investigate the relationship between the order and prediction accuracy, strength prediction models with different orders (second–ninth) were built and compared. Fifty sample points were randomly selected using Latin hypercube sampling, and the prediction accuracy of UCPM was evaluated by examining the fit of the sample points.

The accuracy of UCPMs of various orders was assessed using error metrics, including relative maximum absolute error (RMAE), relative average absolute error (RAAE), and multiple correlation coefficient R^2^. The higher the R^2^ value was, the lower the RMAE and RAAE values were, indicating greater model accuracy. The actual values were calculated using the finite element method, while the predictive values were obtained through the UCPM. Figure 11 demonstrates that a low-order polynomial (second order) is inflexible and cannot accurately fit the details of the response function, resulting in underfitting issues. However, as the order increases from third to seventh, the fit improves, and the prediction accuracy increases. If the order of the Chebyshev polynomials is too high (eighth–ninth order), they may oscillate violently at certain intervals, causing overfitting problems and leading to a decrease in prediction accuracy. Therefore, it is crucial to select the appropriate order.

Table 6 shows that the seventh-order UCPM requires only 32 sample points when there are four independent variables, significantly improving the computational efficiency compared to the traditional prediction model. The UCPM can be applied to rapidly and accurately predict the tensile strength of composites with resin-missing defects.

## 6. Conclusions

A novel UCPM was established by combining the DRM and Chebyshev polynomials to predict the strength of carbon/epoxy composites with resin-missing defects. The relevant conclusions drawn through the research for this paper are as follows: The maximum error between the finite element calculations and experimental results was 3.06%. These results demonstrate that the finite element calculation method is reasonable and feasible. Directly fitting the response function using a Chebyshev polynomial approximation model resulted in an exponential increase in the number of required sample points, leading to significant computational costs. The UCPM model showed better computational efficiency than the conventional Chebyshev polynomial fitting model. The validity of the model was confirmed by comparing it to the experimental results, demonstrating a high prediction accuracy with a maximum prediction error of only 5.20%. The effect of order on UCPMs was studied, and it was found that a low order (second order) resulted in underfitting. Increasing the order (third–seventh order) improved the prediction accuracy of the UCPM. However, if the order is too high (eighth–ninth order), overfitting may occur, leading to a decrease in the prediction accuracy. The UCPM can be applied to rapidly and accurately predict the tensile strength of composites with resin-missing defects.

## Figures and Tables

**Figure 1 polymers-16-00742-f001:**
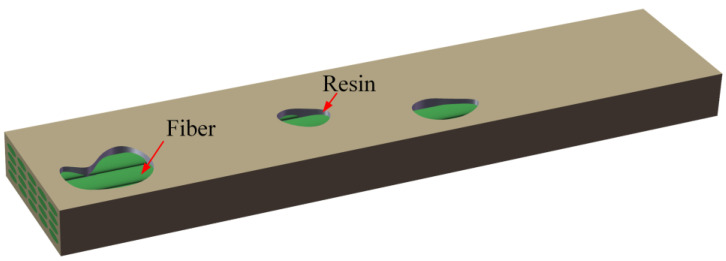
Schematic illustration of resin-missing defects.

**Figure 2 polymers-16-00742-f002:**
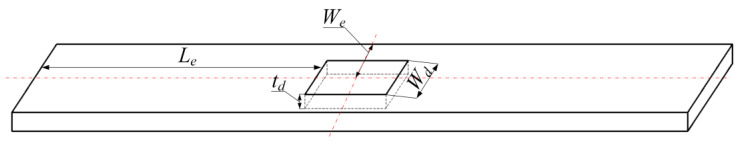
Schematic illustration of composite laminate.

**Figure 3 polymers-16-00742-f003:**
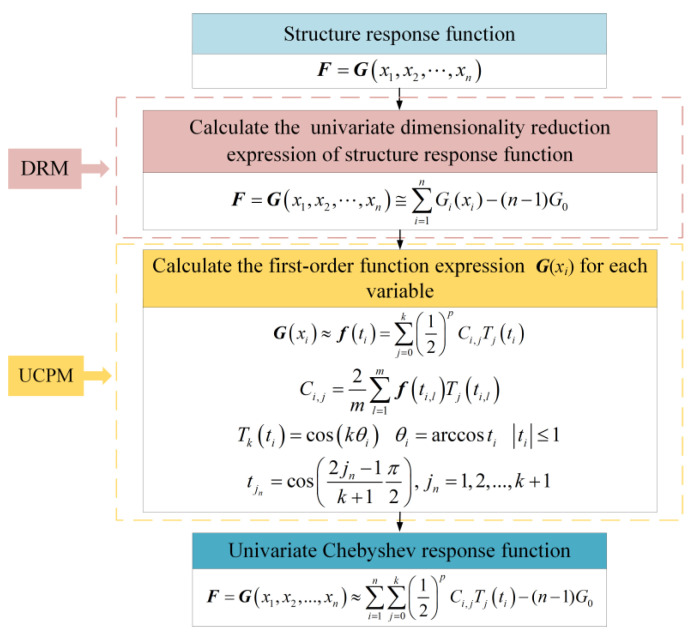
The construction process of the UCPM.

**Figure 4 polymers-16-00742-f004:**
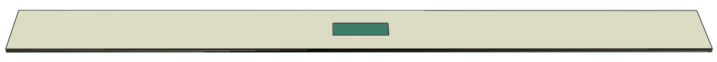
Finite element model.

**Figure 5 polymers-16-00742-f005:**
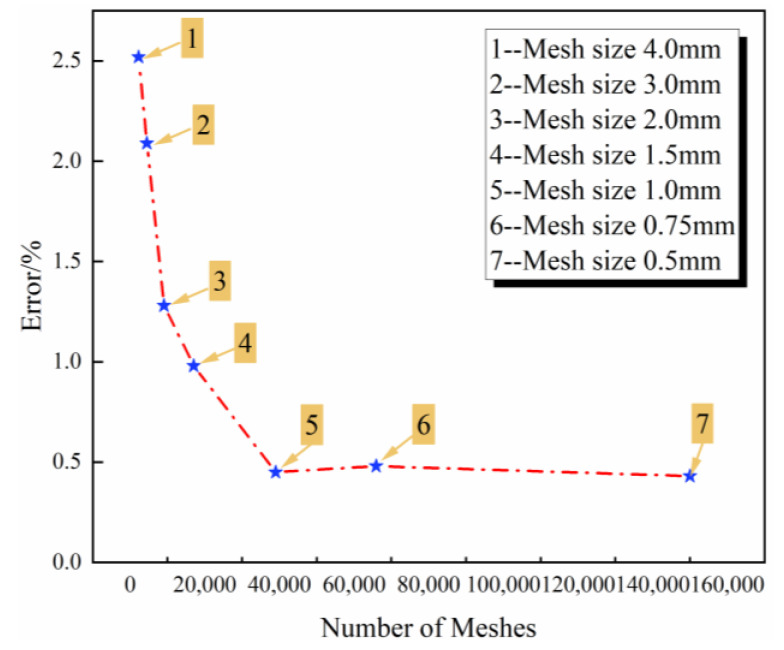
The results of the mesh independence test.

**Figure 6 polymers-16-00742-f006:**
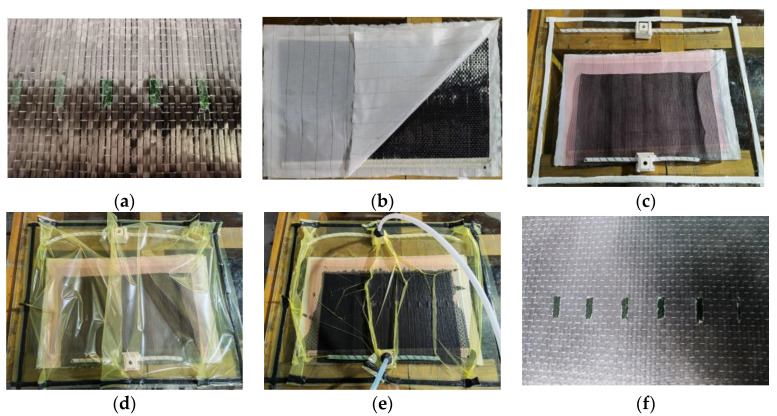
Preparation process of composite laminates. (**a**) Preparation defects. (**b**) Cloth fiber. (**c**) Process accessories. (**d**) Vacuum Sealing. (**e**) Injecting resin. (**f**) Composite laminate.

**Figure 7 polymers-16-00742-f007:**
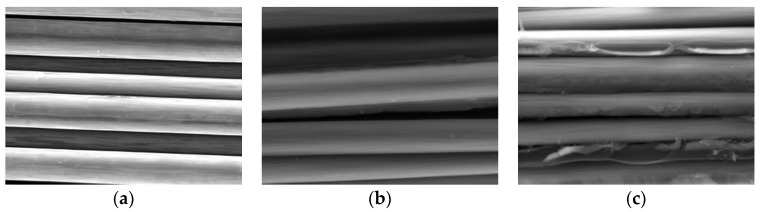
SEM test results. (**a**) Fiber. (**b**) Resin-missing defect area. (**c**) No defect area.

**Figure 8 polymers-16-00742-f008:**
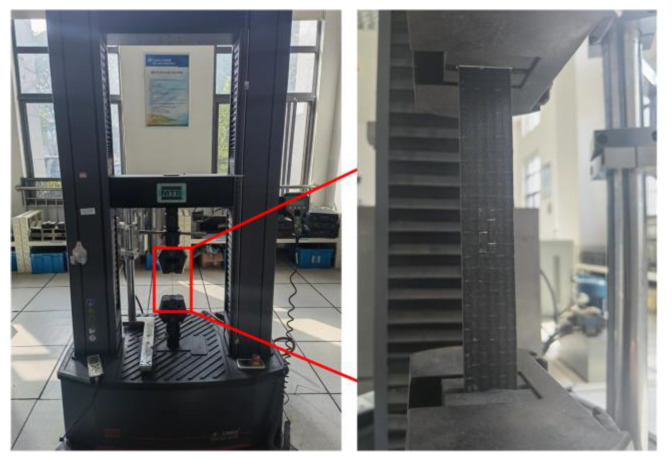
Experimental device.

**Figure 9 polymers-16-00742-f009:**
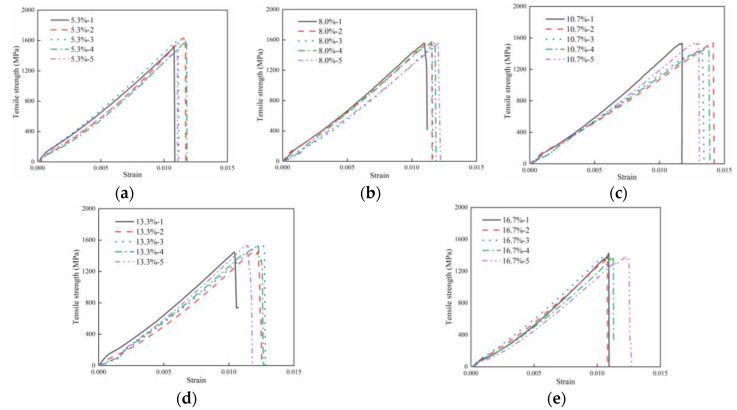
The experimental results. (**a**) Resin-missing defect of 5.3%. (**b**) Resin-missing defect of 8.0%. (**c**) Resin-missing defect of 10.7%. (**d**) Resin-missing defect of 13.3%. (**e**) Resin-missing defect of 16.7%.

**Figure 10 polymers-16-00742-f010:**
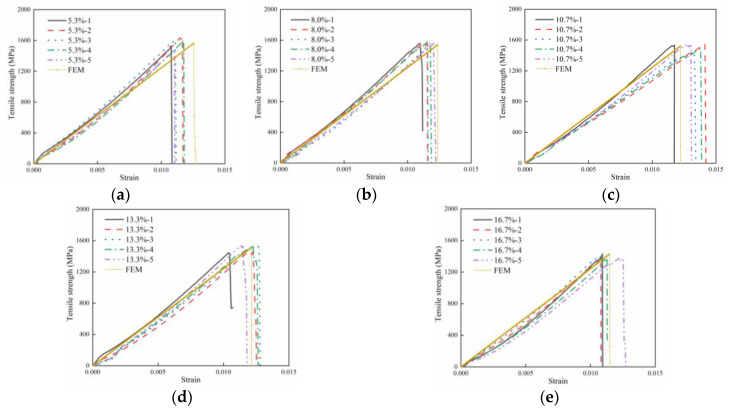
The FEM results. (**a**) Resin-missing defect of 5.3%. (**b**) Resin-missing defect of 8.0%. (**c**) Resin-missing defect of 10.7%. (**d**) Resin-missing defect of 13.3%. (**e**) Resin-missing defect of 16.7%.

**Figure 11 polymers-16-00742-f011:**
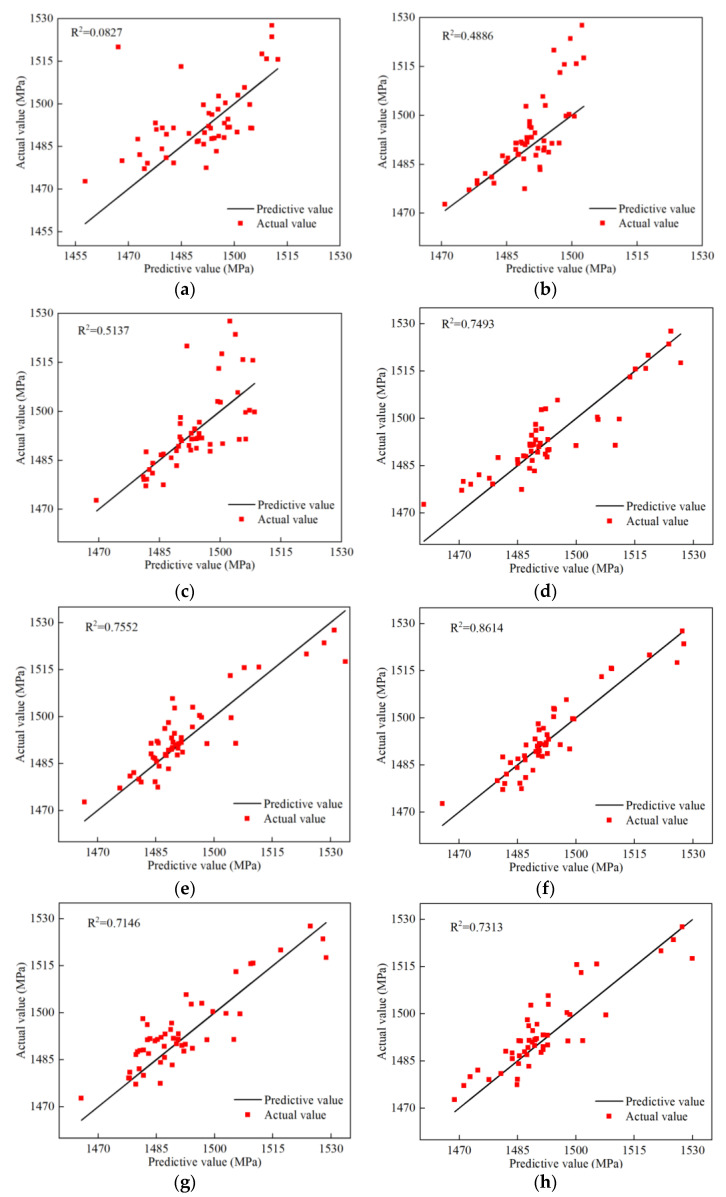
Regression analysis of different orders of UCPM. (**a**) 2nd order. (**b**) 3rd order. (**c**) 4th order. (**d**) 5th order. (**e**) 6th order. (**f**) 7th order. (**g**) 8th order. (**h**) 9th order.

**Table 1 polymers-16-00742-t001:** Range of design variables.

Variables	Minimum/mm	Maximum/mm
*L_e_*	0	115
*W_e_*	2	12.5
*W_d_*	4	25
*t_d_*	0.225	1.35

**Table 2 polymers-16-00742-t002:** Laminate material properties.

Elastic Modulus/GPa		Poisson’s Ratio		Shear Modulus/GPa	
*E* _11_	*E*_22_ = *E*_33_	*μ*_12_ = *μ*_13_	*μ* _23_	*G*_12_ = *G*_13_	*G* _23_
125	8.193	0.3	0.4	3.307	3.151
**Tensile Strength** **/MPa**		**Comprehensive Strength** **/MPa**		**Shear Strength** **/MPa**	
*X* _T_	*Y*_T_ = *Z*_T_	*X* _C_	*Y*_C_ = *Z*_C_	*S*_12_ = *S*_13_	*S* _23_
1630	25	592	98	53	38

**Table 3 polymers-16-00742-t003:** Defect properties.

Elastic Modulus (GPa)		Poisson’s Ratio		Tensile Strength (MPa)	
*E* _11_	*E*_22_ = *E*_33_	*μ*_12_ = *μ*_13_	*μ* _23_	*X* _T_	*Y*_T_ = *Z*_T_
95.88	1 × 10^−5^	0.3	0.2	1367	1 × 10^−5^

**Table 4 polymers-16-00742-t004:** Comparison of results.

Resin-Missing Defect	5.3%	8.0%	10.7%	13.3%	16.7%
Experimental (MPa)	1570.46	1567.56	1509.19	1495.71	1384.72
FEM (MPa)	1563.34	1539.09	1522.7	1511.94	1427.11
Error (%)	0.45	1.82	0.90	1.09	3.06

**Table 5 polymers-16-00742-t005:** Comparison of experimental and predicted results.

	Defect— 5.3%	Defect— 8.0%	Defect— 10.7%	Defect— 13.3%	Defect— 16.7%
Experimental/MPa	1570.46	1567.56	1509.19	1495.71	1384.72
Prediction/MPa	1499.81	1490.27	1485.93	1474.36	1456.79
Error/%	4.50	4.93	1.54	1.43	5.20

**Table 6 polymers-16-00742-t006:** UCPM accuracy comparison of different orders.

Order	2nd	3rd	4th	5th	6th	7th	8th	9th
R^2^	0.0827	0.4886	0.5137	0.7493	0.7552	0.8614	0.7146	0.7313
RMAE	4.1931	3.3212	3.2529	1.3063	1.2222	0.7363	1.2374	1.1873
RAAE	0.0180	0.0130	0.0127	0.0103	0.0098	0.0076	0.0118	0.0107
Sample points	12	16	20	24	28	32	36	40

## Data Availability

Data are contained within the article.

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
