# Peer review of "A Fast and Efficient Approach to Strength Prediction for Carbon/Epoxy Composites with Resin-Missing Defects"

_polymers, 2024, doi:10.3390/polym16060742_

Round 1
Reviewer 1 Report
Comments and Suggestions for Authors
Expand VASI in abstract as vacuum-assisted resin infusion
In Fig.1, how did you distinguish resin and fiber? Better description and illustration is appreciated.
The explanation from Line 127 is not clear and understandable. Did you measure the resin-missing defect (mentioned as 250×25×1.35mm3 in Line 127) or giving an example to explain the method? Add and describe the measurement procedure, if done.
Mesh independent test is much important for FE model confirmation. Hence, I ask authors to add and detail it, add the evidence of mesh independent test into the paper.
Add the C3D8R element details and also reason for choosing it. Add how this element suits the current FE problem.
Table 2 and Table 3 shows the material properties, Where did these data take from? Are they experimentally found or cited from other paper?
I wonder to know how material property of defects were found. Add the details of all these.
To my perspective, Fig. 5 is no needed.
To my understanding, FE analysis and prediction model are continuous and top-down approach. Transformation of FE model data to the prediction model is critical to be discussed in the paper. Hence I ask authors to elaborate the data transformation and integration of FE process with prediction model.
For validation, Samples were fabricated using VASI method. how did you create the defects? Did you purposely missed out resin and created defects? Elaborate.
I advise authors to show micrographs of all samples and explain how defects were.
How did you measure the resin missing defects, elaborate it. Also show the repeatability of the measurement.
Results and discussion section needs a lot improvement. Sections may be re organised in such a way (1) FE analysis (2) Prediction model development (3) Fabrication of samples and computation of mechanical properties (4) Comparison of experimental results with prediction model results.
Prediction model must be detailed and compared with past published articles.
Section 5.9 shows FE results, Are they results obtained during the validation or results obtained before validation of prediction model?
Overall comment:
Papers must be reorganised and sections must be edited. The current form the paper looks very vague. Most importantly authors must address how this research is important to the research community and industries.
Reviewer 2 Report
Comments and Suggestions for Authors
It would be nice to see embedded damage along with surface damage to see the effect of this.
Why was the specific damage area chosen and what is the basis of it?
How does the damage relate to actual components?
L29 should read many composites rather than just composites
L224 How are the defect properties derived?
L234 What is the purpose of FIg5
L245 Does the cloth affect the behavior? Why not use UD prepreg? For a cloth E22 does not equal E33
Is epoxy Vinyl the correct term?
L274 stress v strain curves show a nonlinear take up which also cause errors.
L286 Why does the 8% defect produce larger discrepancies?
Conclusion should include variability in the samples produced.
Round 2
Reviewer 1 Report
Comments and Suggestions for Authors
Appreciated amending the comments.